

# Doppler W-band polarization diversity spaceborne radar simulator for wind studies

Alessandro Battaglia[1,2], Ranvir Dhillon[1], and Anthony Illingworth[3]

[1]University of Leicester, UK
[2]National Centre for Earth Observation, UK
[3]University of Reading, UK

**Correspondence:** Alessandro Battaglia
ab474@le.ac.uk

**Abstract.** CloudSat observations are used in combination with collocated ECMWF wind reanalysis to simulate spaceborne W-band Doppler observations from slant-looking radars. The simulator also includes cross-polarization effects which are relevant if the Doppler velocities are derived from polarization diversity pulse pair correlation. A specific conically scanning radar configuration ("WIVERN"), recently proposed to the ESA-Earth Explorer 10 call that aims to provide global in-cloud winds
for data assimilation, is analysed in detail in this study.

One hundred granules of CloudSat data are exploited to investigate the impact on Doppler velocity estimates from three specific effects: (1) non-uniform beam filling, (2) wind shear, and (3) cross talk between orthogonal polarization channels induced by hydrometeors and surface targets. Errors associated with non-uniform beam filling constitute the most important source of error and can account for almost 1 m s$^{-1}$ standard deviation, but this can be reduced effectively to less than 0.5 m s$^{-1}$
by adopting corrections based on estimates of vertical reflectivity gradients. Wind-shear-induced errors are generally much smaller ($\sim$0.2 m s$^{-1}$). A methodology for correcting such errors has been developed based on estimates of the vertical wind shear and the reflectivity gradient. Low signal-to-noise ratios lead to higher random errors (especially in winds) and therefore the correction (particularly the one related to the wind-shear induced error) is less effective at low signal-to-noise ratio. Both errors can be underestimated in our model because the CloudSat data do not fully sample the spatial variability of the reflectivity
fields whereas the ECMWF reanalysis may have smoother velocity fields than in reality (e.g. they underestimate vertical wind shear).

The simulator allows quantification of the average number of accurate measurements that could be gathered by the Doppler radar for each polar orbit, which is strongly impacted by the selection of the polarization diversity $H - V$ pulse separation, $T_{hv}$. For WIVERN a selection close to 20 $\mu$s (with a corresponding folding velocity equal to 40 m s$^{-1}$) seems to achieve the
right balance between maximizing the number of accurate wind measurements (exceeding 10% of the time at any particular level in the mid-troposphere), and minimizing aliasing effects in the presence of high winds.

The study lays the foundation for future studies towards a thorough assessment of the performance of polar orbiting wide-swath W-band Doppler radars on a global scale. The next generation of scanning cloud radar systems and reanalyses with improved resolution will enable full capture of the spatial variability of the cloud reflectivity and the in-cloud wind fields, thus
refining the results of this study.





## 1 Introduction

Observation of atmospheric 3D winds and their monitoring at multiple temporal and spatial scales has been identified as a priority in the recent NASA Decadal Survey (The Decadal Survey, 2017). Large-scale winds are paramount in the transport of energy and water through the atmosphere and, together with vertical motions of convection, are the main driver in controlling

water vapour transport around the globe. They are an essential element in the circulation of the atmosphere, in coupling clouds and the general circulation, in understanding the hydrological cycle and in untangling climate challenges (Bony et al., 2015).

Zeng et al. (2016) state that "it is important to avoid all-or-nothing strategies for three-dimensional (3D) wind vector measurements", i.e. that progress can be achieved with observing strategies that are not comprehensive - e.g. are only effective in certain conditions (clear sky, cloudy, etc.) and maybe capable of measuring only one or two components of the wind- and

that complement each other. An integrated approach of active sensing (lidar, radar, scatterometer) and passive imagery or radiometry-based atmospheric motion vectors is therefore envisaged for improving global observations of winds in the future. In this synergistic approach active sensors on LEO satellites could be used to calibrate observations from geostationary satellites that have excellent temporal coverage but are affected by large errors in assigning a height to the retrieved wind. Profiles of tropospheric winds currently have the highest priority ("the holy grail") for all operational weather agencies. Doppler active

sensors (lidars and radars) on LEO satellites which use atmospheric targets as wind tracers are unanimously credited to be the key instruments to achieve this priority. While an explorer/incubation mission of this type is recommended by NASA for the next decade (The Decadal Survey, 2017), the ESA Earth Explorer programme already has two missions in the pipeline aiming towards this goal. The ESA Aeolus mission to be launched in mid-2018 (Stoffelen et al., 2005) will provide the first Doppler lidar measurements of the line-of-sight winds in clear air and thin ice clouds. It will be followed by the ESA-JAXA Earth-

CARE mission (launch planned for 2020, Illingworth et al. (2015)) that will provide vertical velocities of cloud particles via a nadir-pointing Doppler W-band radar. For the first time sedimentation rates of ice crystals (Kollias et al., 2014) and convective up- and down-draughts (Battaglia et al., 2011) will be observed from space.

However, none of these missions will be able to provide horizontal winds in deep cloud systems. McNally (2002) showed that "sensitive" areas where observations have the largest potential to improve forecasts are often cloudy. The nexus between

clouds and winds is revealed in Fig. 1, a snapshot of the global winds and ice water content (IWC) at a height of 8 km from the ECMWF model for 12 noon on 12 January 2018. Particularly obvious are the high values of IWC and rapidly changing winds associated with the storm to the south of Japan, as is the case for other mid-latitude depressions. Areas where winds change rapidly are often associated with clouds, and only radars can penetrate such areas.

Recent European Space Agency (ESA)-funded studies suggested addressing this wind observational gap by using W-band

radars - ideal for their sensitivity and narrow beamwidths - with scanning and Doppler capabilities. Both a stereoradar and a conically scanning configuration have been proposed (Battaglia and Kollias, 2014a; Illingworth et al., 2018a). The former investigates the link between microphysical and dynamical structures of cloud systems, including convective systems, while the latter, known as WInd VElocity Radar Nephoscope (WIVERN), aims to provide global in-cloud winds for data assimilation and has now been proposed to the ESA Earth Explorer 10 call (Illingworth et al., 2018b). By conically scanning an 800 km





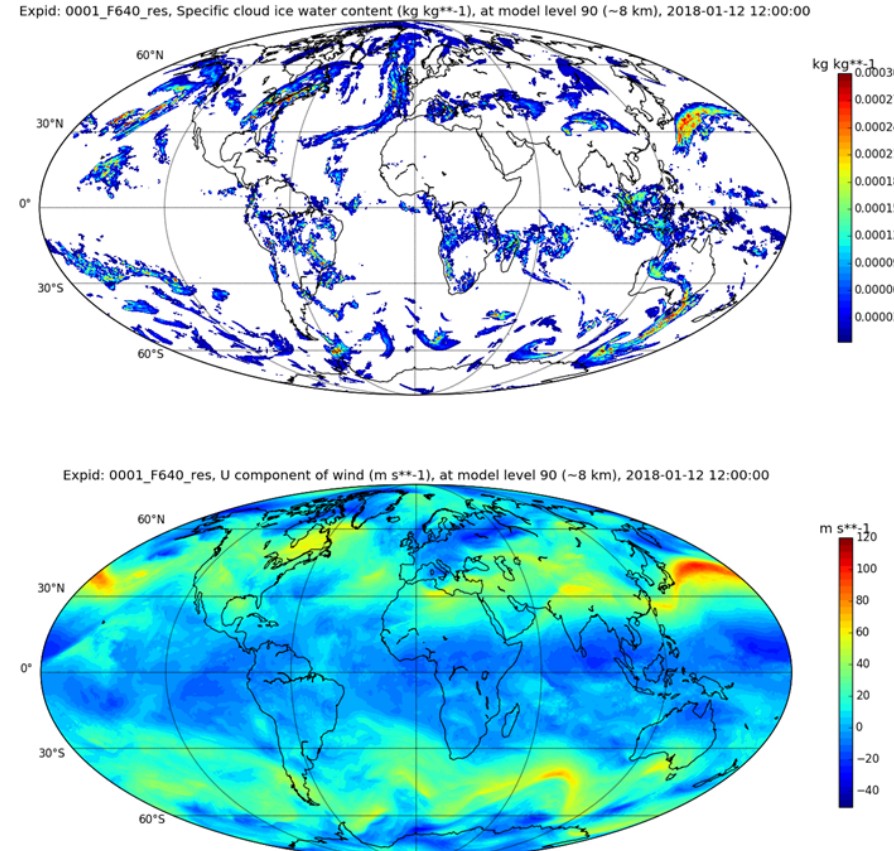

**Figure 1.** Cloud ice water content (top) and ECMWF zonal winds (bottom) and at a height of 8 km at noon on 2018-01-02. The data are plotted at a resolution of about 14 km. Ice water is only plotted when the mixing ratio exceeds $10^{-6}$ kg kg$^{-1}$ or $\sim 2 \times 10^{-3}$ gm$^{-3}$ or a reflectivity of $\sim$-23 dBZ (Hogan et al., 2006). Plot courtesy of M. Rennie, ECMWF.

wide ground track the radar allows the measurement of large-scale winds associated with long-lived systems to be assimilated into weather forecast models with daily coverage at mid-latitudes.

The commonalities of both systems are:

– they look/scan at a slant view (incidence angles in the range between 40° to 50°) in order to capture horizontal winds;

– they adopt polarization diversity to overcome the range-Doppler dilemma and to cope with the short decorrelation times associated with the Doppler fading inherent to millimetre Doppler radars on fast-moving low Earth orbiting satellites (Tanelli et al., 2002; Kollias et al., 2014);

– they require large antennas to optimize the Doppler quality.





By focusing on slant-looking Doppler radars adopting polarization diversity this study aims to define a simulation framework which enables the assessment of radar performance on a global scale.

Doppler velocity accuracy requirements depend on the application but the WMO requirement for assimilating winds is to have errors lower than 2 m s$^{-1}$ at a horizontal sampling of 50 km and a vertical resolution of 1 km or better (see Illingworth et al. (2018a) for a thorough discussion of wind user requirements). Noticeably Horanyi et al. (2014) state that assimilating winds with biases of 1-2 m s$^{-1}$ can actually degrade the forecast. It is therefore important to assess the accuracy and precision of Doppler velocities for future spaceborne wind-observing radars. There are several sources of uncertainty associated with polarization diversity Doppler measurements from space, such as errors linked to non-uniform beam filling (NUBF) (Tanelli et al., 2002) coexisting with or without wind shear, cross talk between the H and V returns (Pazmany et al., 1999; Illingworth et al., 2018a; Wolde et al., 2018), clutter contamination, aliasing (Battaglia et al., 2013; Sy et al., 2013), mispointing (Tanelli et al., 2005; Battaglia and Kollias, 2014b), multiple scattering (Battaglia and Tanelli, 2011) and errors related to the Doppler estimators in the pulse-pair processing. The uncertainties associated with the pulse-pair processing are very well characterized: they depend on the signal-to-noise ratio (SNR), the radar Doppler spectral width, and the number of averaged samples (Doviak and Zrnić, 1993; Battaglia et al., 2013; Illingworth et al., 2018a). The other errors are more complicated and are generally assessed via simulations of cloud-resolving model scenes (Battaglia et al., 2013; Leinonen et al., 2015). In other cases, in order to avoid the uncertainties associated with transforming bulk microphysical properties to radar reflectivities, ground-based (Kollias et al., 2014; Burns et al., 2016) or aircraft (Sy et al., 2013, 2014) observations at the same radar frequency are exploited, with the advantage of reproducing naturally observed fields of reflectivity, together with their spatial variability. On the other hand such observations are seldom representative of the global scale. In addition only Battaglia et al. (2013) and Battaglia and Kollias (2014a) have addressed issues related to polarization diversity in a simulation framework.

In this study we exploit CloudSat W-band radar observations in combination with collocated ECMWF wind reanalysis to simulate spaceborne W-band Doppler radar observations from off-nadir scanning radars. The simulator also includes cross-polarization induced by atmospheric targets in order to assess its impact on Doppler polarization diversity pulse-pair estimates. The proposed simulation framework can therefore enable an error budget assessment on a global scale for a satellite mission adopting a sun-synchronous orbit similar to CloudSat. Sect. 2 provides infomation about the datasets that have been used while Sect. 3 describes the Doppler radar simulator and its application to the case study of hurricane Igor. In Sect. 4 $\tilde{1}00$ orbits of CloudSat data are exploited to characterize the performance of a W-band Doppler system on a global scale. Conclusions and future work are presented in Sect. 5.

## 2 Datasets

In order to simulate realistic scenes for assessing the capabilities of future spaceborne W-band Doppler radars two ingredients are needed: (1) W-band reflectivity profiles through a variety of cloud regimes with spatial resolutions comparable or better than those to be simulated and capable of representing the natural variability, and (2) wind profiles for the whole troposphere. In this work the former are taken from the CloudSat W-band radar, and the latter are extracted from ECMWF products.



## 2.1 CloudSat products

The CloudSat 94 GHz (3.2 mm) Cloud Profiling Radar (CPR) measures reflectivities from cloud- and precipitation-sized particles at a vertical resolution of 480 m for a cross-track/along-track horizontal footprint of 1.5 km×2.5 km (Stephens et al., 2008). The radar has been collecting data on a polar sun-synchronous orbit since its launch in 2006 (Tanelli et al., 2008). This study makes use of different Level 2 CloudSat data products: the 2B-GEOPROF (Mace et al., 2007), the 2B-CLDCLASS-LIDAR (Sassen et al., 2008) and the 2C-RAIN-PROFILE (Haynes et al., 2009) (more details at http://www.cloudsat.cira.colostate.edu/).

## 2.2 ECMWF product

The European Centre for Medium-Range Weather Forecasts (ECMWF) maintains an archive of meteorological and air-quality data - covering a wide range of parameters including, for example, gas and pollutant concentrations, precipitation measurements and wind values - central to its core mission of producing numerical weather forecasts and monitoring the Earth system (https://www.ecmwf.int/).

For the study covered here, ECMWF global wind fields (u- and v-components) at a temporal resolution of 6 hours and latitude and longitude resolutions of $0.1°$ are used. The wind fields are provided at 25 pressure levels ranging from 1 to 1000 hPa. They are collocated with the CloudSat measurements by selecting the nearest ECMWF grid point (latitude and longitude) to the CloudSat position and by temporally interpolating to each CloudSat profile time stamp between the two closest ECMWF time stamps.

## 3 Simulator

### 3.1 Radar configuration

The radar configuration that will be used throughout this paper is the one depicted in Fig. 2. The radar specifications are detailed in Tab. 1. The configuration corresponds to the one proposed for the WIVERN mission (which involves a conically scanning system) when the antenna is looking in the same direction as the spacecraft motion (Illingworth et al., 2018a, b) and is very similar to the one proposed in Battaglia and Kollias (2014a). Therefore we will refer to it as the "WIVERN forward" configuration. The antenna pattern is assumed Gaussian with a two-way gain equal to:

$$G^2(\theta) \quad = \quad G_0^2 \exp\left[-8\,log(2)\left(\frac{\theta}{\theta_{3dB}}\right)^2\right], \tag{1}$$

where $G_0$ is the antenna gain in the boresight direction, $\theta$ is the antenna polar angle with respect to the boresight and $\theta_{3dB}$ is the antenna 3-dB beamwidth. The Doppler velocity is computed as:

$$v_D(r) = \frac{\iiint_V v_r\,Z\,G^2 dV}{\iiint_V Z\,G^2 dV} \tag{2}$$

where $v_r$ is the wind velocity along the line of sight. Since CloudSat only provides a 2D-curtain through cloud systems, the integral in Eq. (2) is reduced to a two-dimensional integral that can be performed in polar coordinates.




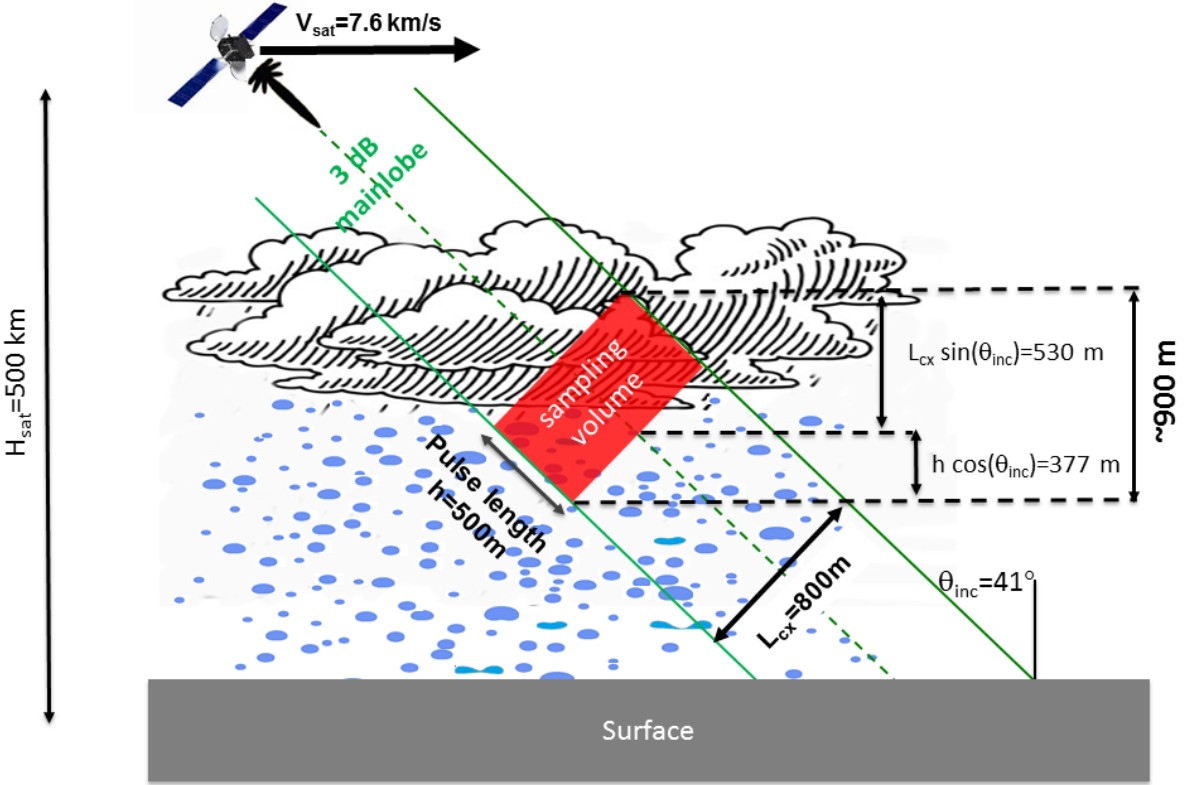

**Figure 2.** Schematic illustrating the geometry of a radar observing cloud and precipitation at a slant angle. The specifics of the radar are detailed in Tab. 1

## 3.2 Case study: Hurricane Igor

The simulator rationale is demonstrated for a case study based on a CloudSat overpass over Hurricane Igor. Hurricane Igor originated from a broad area of low pressure that moved off the Cape Verde islands on September 6, 2010. It subsequently developed into a tropical depression on September 8 and reached Category 4 status on September 12 with winds peaking at $70\ \mathrm{m\ s^{-1}}$.

5    CloudSat in ascending orbit overpassed to the East of the storm centre on September 16 (see http://cloudsat.atmos.colostate.edu/news/2010_ when Igor had gradually started weakening. Fig. 3 shows the CloudSat radar vertical reflectivity profile derived from the 2B-GEOPROF product across the whole hurricane: the system extends for about 2200 km horizontally with clouds towering to almost 16 km. Some deep convection and heavy precipitation are clearly present close to the centre of the plot corresponding to the eye wall: attenuation is so strong that even the surface signal is completely attenuated. Some deep isolated convective



**Table 1.** Specifics of the radar for the simulation. The configuration here adopted is the one proposed for WIVERN in a recent ESA Earth Explorer 10 call.

| | |
|---|---|
| Satellite altitude, $h_{sat}$ | 500 km |
| Satellite velocity, $v_{sat}$ | 7600 ms$^{-1}$ |
| Incidence angle, $\theta_i$ | 41° |
| RF output frequency | 94.05 GHz |
| Pulse width | 3.3 $\mu$s |
| Antenna beamwidth, $\theta_{3dB}$ | 0.07° |
| Transmit polarization | H or V |
| Cross-polarization | <-25 dB |
| Single pulse sensitivity | -19 dBZ |
| H-V Pair Repetition Frequency | 4 kHz |
| Footprint speed | 300 kms$^{-1}$ |
| Number of H-V Pairs per 1 km integration length | 10 |

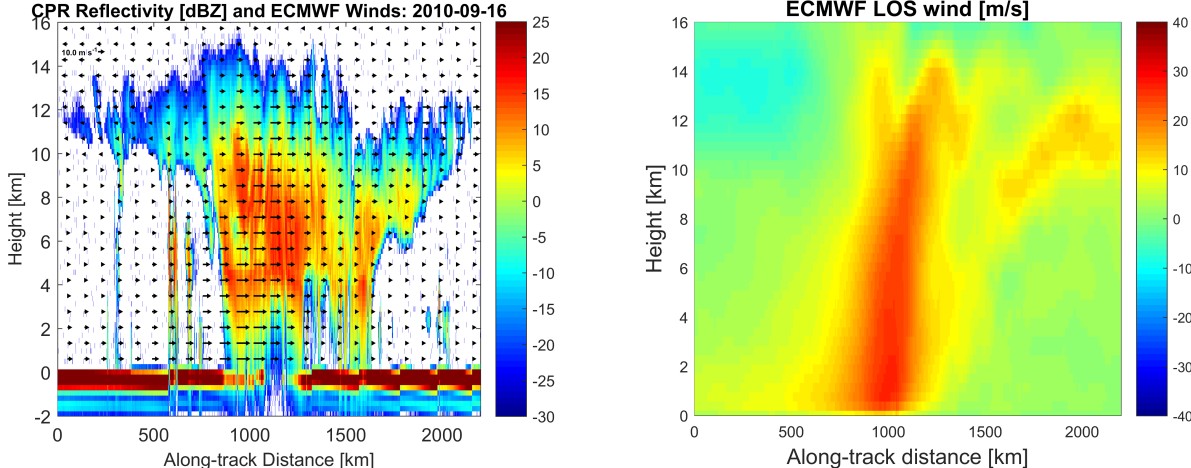

**Figure 3.** Left panel: CloudSat radar vertical reflectivity profile at W-band for overpass of Atlantic Hurricane Igor on 16 Sept 2010 between 1713 and 1728 UTC (corresponding to along-track distance of 2200 km). The ECMWF line-of-sight winds projected onto the CloudSat curtain are plotted as arrows. Right panel: Contour plot of ECMWF line-of-sight winds. The reflectivity data are provided at Cloudsat resolution (1.1 km horizontal, 0.5 km vertical) whereas the ECMWF winds are at 0.1° resolution (≈10 km). For presentation purposes only winds every 10 km along the curtain and every 700 m in the vertical are shown (left panel).

towers are present to the South (left side of the panel) in association with the spiral bands. The cirrus canopy stretches across most of the overpass, but it is much taller in the Southern part.





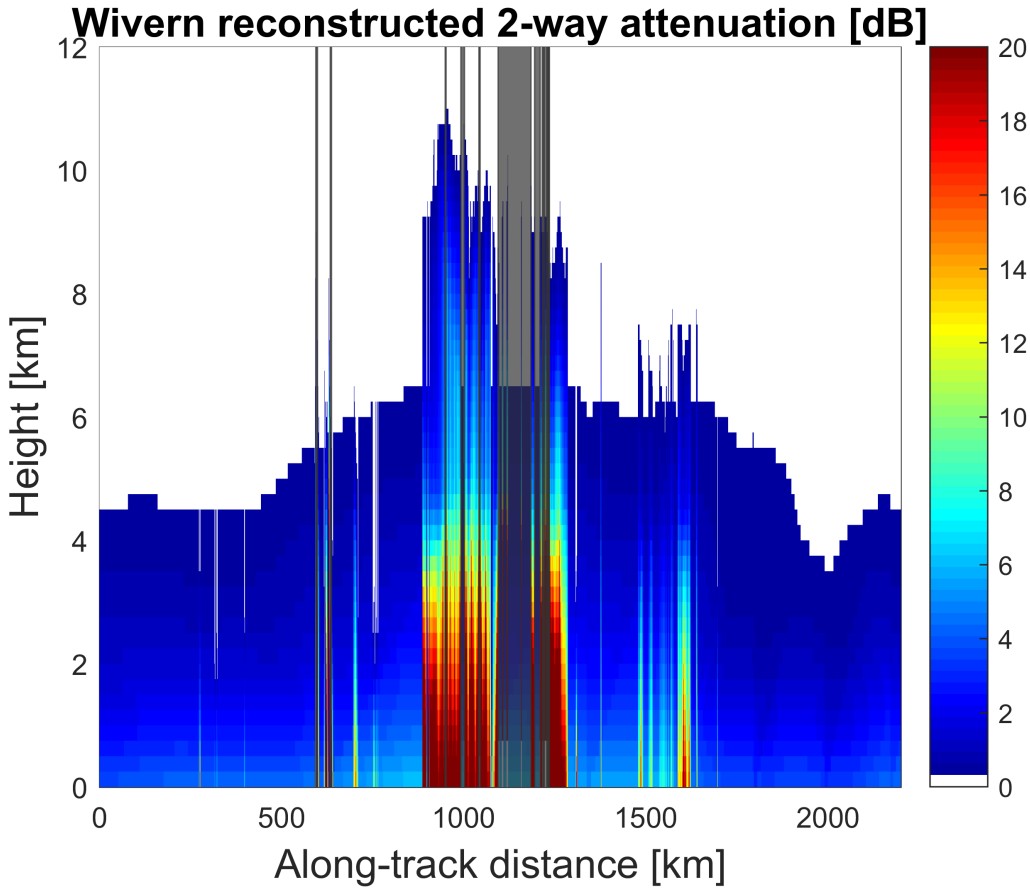

**Figure 4.** Two-way attenuation as retrieved in the 2C-RAIN product for the scene shown in Fig. 3. Grey bands correspond to profiles where the retrieval in the 2C-RAIN product is not applicable. Data are provided at Cloudsat resolution (1.1 km horizontal, 0.5 km vertical).

The wind field of course varies appreciably over the scene, and this is exemplified by the characteristic change in the line-of-sight velocity component that occurs across the hurricane (indicated by the arrows in Fig. 3). This line-of-sight component, which is determined by combining ECMWF and CloudSat data, varies significantly over the horizontal extent of the hurricane, with values ranging from approximately -10 to +30 m s$^{-1}$ at along-track distances of about 0 and 1000 km respectively (Fig. 3).

5    The region with the highest wind magnitudes is associated with the upper levels in the central tallest part of the system.

The 2C-RAIN product reconstructs vertical profiles of attenuation (see Fig. 4) and of effective reflectivities based on an optimal estimation framework. The retrieval is not applicable in regions of strong convection in the presence of multiple scattering and high attenuation (Matrosov et al., 2008; Battaglia et al., 2011) where no convergence of the algorithm is obtained. These regions are marked by the grey stripes in Fig. 4. No reconstruction of simulated profiles will be attempted in such regions.

10    Note that in regions filled with precipitation the 2-way path-integrated attenuation can exceed 30 dB.





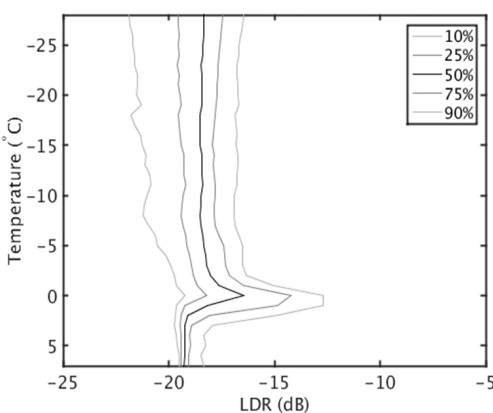

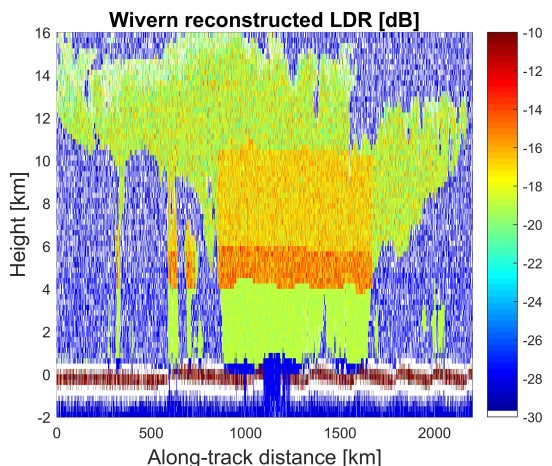

**Figure 5.** Left panel: climatological observations of W-band $LDR$ at the Chilbolton observatory for an elevation angle of 45° (courtesy of John Nicol, University of Reading). Right panel: simulated $LDR$ for the scene shown in Fig. 3 at Cloudsat resolution (1.1 km horizontal, 0.5 km vertical).

In order to simulate a Doppler radar with polarization diversity linear depolarization ratio ($LDR$) profiles are also needed (see discussion later in Sect. 3.3.3). A crude $LDR$ is reconstructed based on climatological observations of $LDR$ at the Chilbolton observatory (see left panel in Fig. 5). Data were collected during June and July 2017 at 45° elevation with the W-band Galileo polarimetric radar. Different hydrometeors (as derived from the 2B-CLDCLASS-LIDAR product) are assigned

$LDR$ values drawn from a normal distribution with 0.5, 1.5, 2 and 1.5 dB standard deviation and mean values of -19, -18, -16 and -17 dB for rain, ice crystals, melting particles and the mixed-phase region, respectively. Surface $LDR$ are assumed to be normally distributed around -14 dB and -7 dB for sea and land respectively with 2 dB standard deviation (Battaglia et al., 2017).

In order to produce realistic simulations of slant-looking radars two aspects must be accounted for.

1. The slant viewing geometry (with an increased cumulative attenuation compared to nadir-looking radar) and the appropriate antenna pattern (Eq. 1) must be included in the integral in Eq. 2; the integration is carried out at the initial 1.1 km integration length of CloudSat. Further along-track averaging is performed later (e.g. resolution of winds for data assimilation is about 20 km).

  2. The ground clutter must be significantly reduced, especially over ocean. In Fig. 3 the ocean surface return is very strong

because CloudSat is almost nadir looking. On the other hand, in the configuration shown in Fig. 2, the surface clutter will be significantly lower. Recent airborne studies at 94 GHz and WIVERN incidence angles (Battaglia et al., 2017) have established the natural variability of the normalized radar surface backscattering, $\sigma_0$. Accordingly in this study, sea



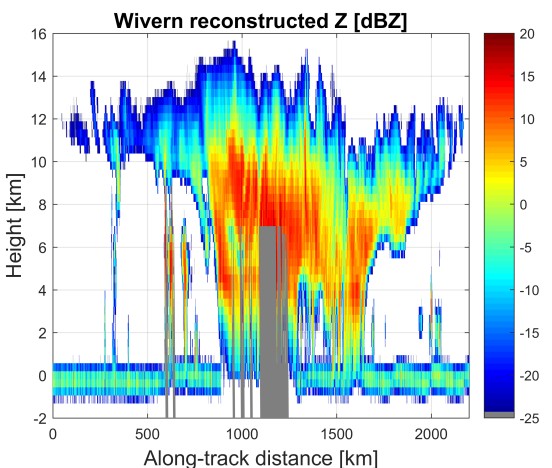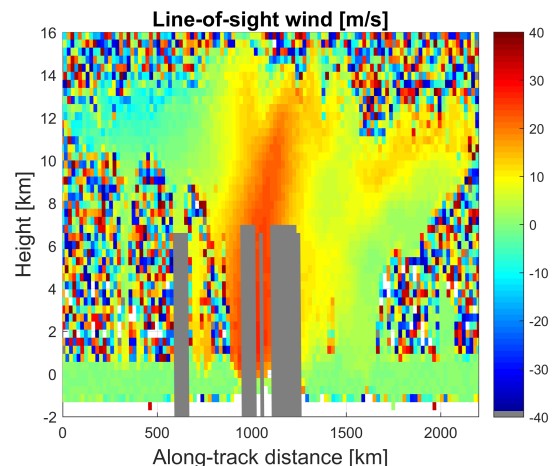

**Figure 6.** Reconstructed reflectivity (left) and line-of-sight Doppler velocities (right) simulated for a system with the specifics listed in Tab. 1 and starting from the CloudSat scene illustrated in Fig. 3. The regions shaded in grey correspond to areas where no reconstruction of the reflectivity profile is possible due to severe attenuation/multiple scattering. The Doppler velocities are produced with a 20 km integration length for a total of 200 $H - V$ pairs with a $T_{hv} = 20~\mu$s (which corresponds to a Nyquist velocity of 40 ms$^{-1}$).

(land) surface $\sigma_0$ values have been assumed to be normally distributed around -25 dB (-8 dB) with standard deviation equal to 5 dB (4 dB).

The result of the simulator is shown in Fig. 6. Note that the grey stripes correspond to the regions where no attenuation correction is deemed possible, which go from approximately 2 km above the freezing level to the ground for the grey stripes of
Fig. 4 corresponding to the grey bands where the 2C-RAIN product is not applicable. Because of the slant geometry the grey bands are now a little bit wider than for Fig. 4.

### 3.3   Errors

The simulation framework is ideal for properly assessing Doppler (reflectivity-weighted) line-of-sight velocity estimate ($\hat{v}_D$) errors on a global scale. Here we will focus our analysis on three different source of errors which are related to the spatial
structure of the wind and of the reflectivity fields: errors due to NUBF, to the presence of wind shear and to the cross talk between channels induced by atmospheric targets when adopting polarization diversity.

#### 3.3.1   Non uniform beam filling: satellite motion-induced biases

For a fast-moving spaceborne Doppler radar, radar reflectivity gradients within the radar sampling volume can introduce a significant source of error in Doppler velocity estimates (Tanelli et al., 2002). In fact the component of the satellite velocity
presents a shear across the backscattering volume. In Fig. 7 the configuration adopted in Fig. 2 is used to illuminate the



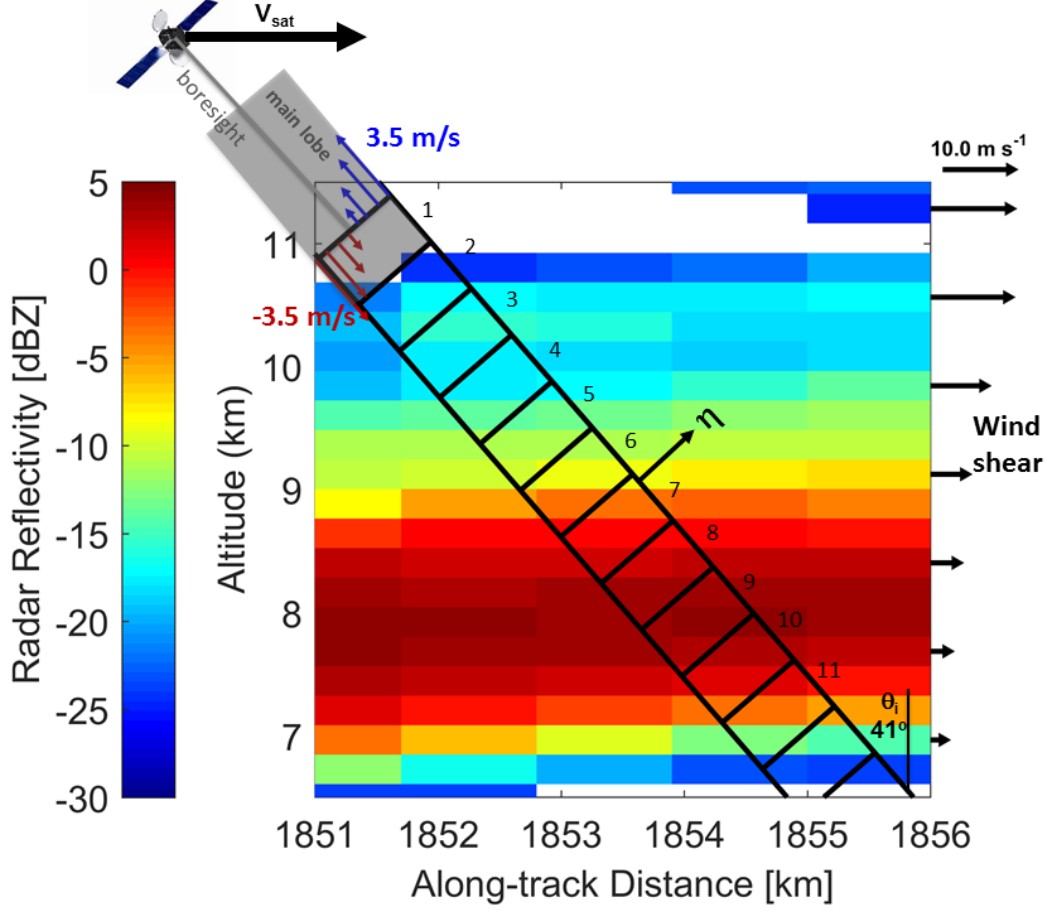

**Figure 7.** Diagram explaining Doppler velocity errors introduced by NUBF. The reflectivity profiles are extracted from hurricane Igor (ice clouds corresponding to the arrow in the bottom left panel of Fig. 3). The black rectangles represent the backscattering volumes associated with the 3-dB antenna main lobe.

small radar volume identified by the black arrow in Fig. 3. If the frequency Doppler shift and the associated Doppler velocity corresponding to the antenna boresight direction are perfectly compensated for and set to zero, then the forward (backward) part of the backscattering volume appears to move upward (downward) as illustrated by the blue (red) arrows. Across the 3-dB footprint size this velocity ranges from -3.5 to +3.5 m s$^{-1}$. When coupled with a reflectivity gradient this satellite-motion-induced velocity shear can produce a bias. For instance for the backscattering volumes labelled as 11 (6) there is a positive (negative) reflectivity vertical gradient which will produce an upward (downward) bias.

An estimate of these biases is obtained by considering the difference between $\hat{v}_D$ and $\hat{v}_D[v_{sat} = 0]$ where the latter is evaluated by setting the satellite velocity to 0 m s$^{-1}$. The NUBF-induced bias corresponding to the scene of hurricane Igor





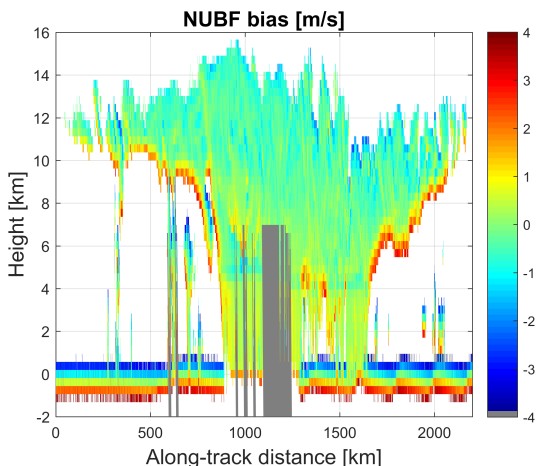 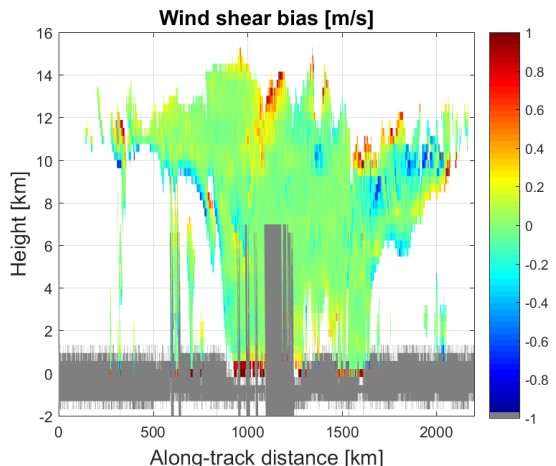

**Figure 8.** NUBF-induced (left panel) and wind-shear-induced (right panel) errors for the hurricane Igor scene shown in Fig. 6. Most of the biases are near the cloud edge; widespread biases of ∼1 m s$^{-1}$ must be avoided.

depicted in Fig. 6 is shown in Fig. 8. The main feature is represented by upward biases at the cloud top and downward biases at the cloud base, both up to several m s$^{-1}$.

For nadir pointing radars notional studies demonstrated that such biases can be mitigated by estimating the along-track reflectivity gradient because NUBF-induced biases are expected to be linearly proportional to such reflectivity gradients (Schutgens, 2008; Kollias et al., 2014; Sy et al., 2013). Similarly in a slant-looking geometry the relevant gradients are those along the direction orthogonal to the boresight and lying in the plane containing the satellite velocity and the antenna boresight direction ($\eta$ direction in Fig. 7, Battaglia and Kollias (2014a)). If conically scanning systems are considered it will be more challenging to retrieve the Z-gradients along such directions for all scanning angles but it is expected that the dominant contribution will come from the vertical reflectivity gradients. In the presence of a reflectivity gradient, $\nabla_z Z$ (in dB m$^{-1}$), if the reflectivity field can be approximated to vary linearly within the backscattering volume then the bias introduced by the satellite motion is equal to (Sy et al., 2013; Battaglia and Kollias, 2014a):

$$\Delta_{satellite\ motion} = v_{sat} \frac{\nabla_z Z\, r}{4.343} \frac{\sin(2\theta_i)}{32\,log(2)} \theta^2_{3dB} \tag{3}$$

where $r$ is the range from the radar. For instance, for the "WIVERN" configuration (Tab. 1) this corresponds to a bias of 0.077 m s$^{-1}$ per dB km$^{-1}$.

### 3.3.2 Wind shear

Similarly biases in the radar-derived winds may arise when there is a vertical wind shear (see arrows on the right hand side of Fig. 7) coupled with a large vertical gradient of radar reflectivity across the radar backscattering volume. An estimate of these biases is obtained by considering the difference between $\hat{v}_D$ and $\hat{v}_{AW}$ where the latter is the line-of-sight velocity estimate





averaged over the antenna pattern but not antenna-weighted, i.e.

$$v_{AW}(r) = \frac{\iiint_V v_r\, G^2 dV}{\iiint_V G^2 dV}.$$

For the Hurricane Igor case study, the results are shown in the right panel of Fig. 8: wind-shear-induced biases are generally smaller than NUBF-induced biases with amplitudes up to 1 m s$^{-1}$ and confined to the areas at the edge of clouds characterized by large wind shear and vertical reflectivity gradients.

Since the vertical wind shear is generally considerably larger than the horizontal one, under the assumption that the reflectivity and wind fields can be approximated to vary linearly within the backscattering volume, the bias due to wind shear can be approximated as:

$$\Delta_{wind\,shear} \quad = \quad v_D - v_{AW} = \frac{\nabla_z Z\, \nabla_z v}{4.343} \left[ \frac{\Delta r^2}{12} \cos^2\theta_i + \frac{r^2 \theta_{3dB}^2}{16\,log(2)} \sin^2\theta_i \right] \qquad (4)$$

where $\nabla_z Z$ and $\nabla_z v$ are the reflectivity and wind vertical gradients expressed in dB m$^{-1}$ and in s$^{-1}$. For instance for the "WIVERN" configuration (Tab. 1) this corresponds to a bias of 0.37 m s$^{-1}$ per dB km$^{-1}$ for a wind shear of 0.01 s$^{-1}$. The reflectivity gradients and wind shear along the vertical direction can be inferred from adjacent gates and therefore a correction can be attempted.

### 3.3.3 Cross-talk

Doppler systems adopting polarization diversity assume that the $V$ and $H$ waves propagate and scatter independently without any interference. In reality the effect of cross-polarization is to produce an interference signal in each co-polar channel depending on the temporal shift between the $H$ and $V$ pair and the strength of the cross-polar power, and appear as "ghost echoes". Cross-talk between the two polarizations can occur either at the hardware level or can be induced by propagation and/or backscattering in the atmosphere. While the former is typically reduced to values lower than -25 dB the latter can be important and is characterized by the $LDR$. The phases of ghost echoes are incoherent with respect to the echoes of interest so do not bias the velocity estimates, but increase their random error as a function of the signal-to-ghost ratio (Pazmany et al., 1999; Wolde et al., 2018). This quantity depends on the reflectivity profile structure and on $T_{hv}$, the time separation between the $H-$ and $V-$ polarized pulses. The full theory is reviewed in depth in Wolde et al. (2018), formulas (6-11). An example of the line-of-sight measured Doppler velocity for a $T_{hv} = 20\ \mu$s is shown in the right panel of Fig. 6 for the "WIVERN forward" configuration. The Doppler winds compare well with those from ECMWF, shown in the right panel of Fig. 3. Note how the measurements become increasingly noisier when moving towards lower SNRs.

## 4 Statistical analysis

Eight days of CloudSat data, from 1 to 8 Sept 2010, have been used to obtain simulated residual errors, defined as the difference between the actual velocity biases and the velocity corrections obtained using the formulae given above. The dataset comprises 94 granules and therefore covers a variety of cloud types with a multitude of characteristics. The CloudSat data granules





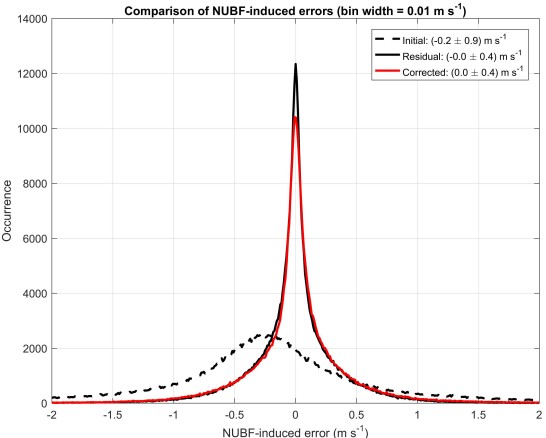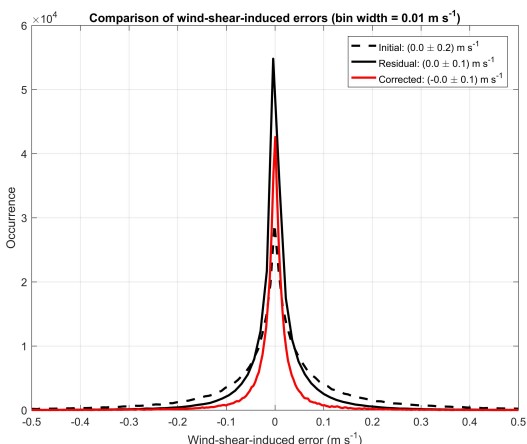

**Figure 9.** Left (right) panel: distribution of 8-day CloudSat dataset NUBF-induced (wind-shear-induced) errors. Black dashed line: initial errors; black solid line: errors after correction with perfect measurements; red line: errors after correction with noisy measurements. A $T_{hv} = 20~\mu$s and a spectral Doppler width of 4 m s$^{-1}$ have been assumed when including noise.

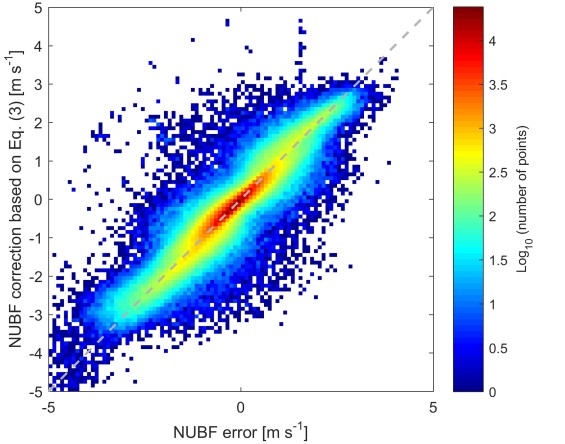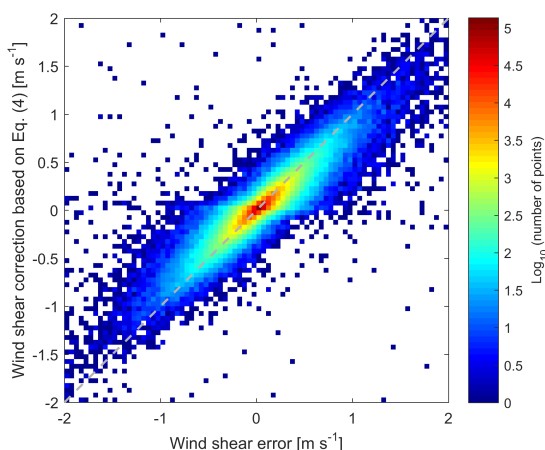

**Figure 10.** Left (right) panel: density plot for the 8-day CloudSat dataset illustrating the NUBF-induced (wind-shear-induced) bias vs bias correction estimated according to Eq. 3 (Eq. 4).

used for this study were chosen based upon whether data files corresponding to the four relevant CloudSat data products (2B-GEOPROF, 2B-CLDCLASS, 2C-RAIN-PROFILE and ECMWF-AUX) were all present simultaneously.





### 4.1 NUBF and wind shear errors and corrections

The technique demonstrated for Hurricane Igor in Sect. 3.2 has been applied to the whole dataset. First the magnitudes of the errors induced by NUBF and wind shear are evaluated, e.g. the wind-shear-induced biases are obtained by subtracting the ECMWF (antenna-weighted) winds from the reflectivity-weighted winds. The black dashed lines in Fig. 9 show the distribution

for the NUBF-induced (left) and wind-shear-induced error (right), with the former significantly larger (standard deviation of 0.9 m s$^{-1}$ vs 0.2 m s$^{-1}$) compared to the latter and slightly biased (-0.30 m s$^{-1}$). This bias is due to the fact that the average reflectivity vertical gradient (when all heights are considered) is slightly negative. Note that this bias will cancel out if a backward look is also adopted (as for the conically scanning WIVERN).

Since CloudSat reflectivity and ECMWF wind fields are defined at coarse horizontal scales the simulation may not account

for the full variability inside the backscattering volume; thus the errors may be underestimated. Second, the goodness of the corrections derived from the formulae given by Eqs. (3-4) is tested by comparing NUBF-induced and wind-shear-induced errors with their corresponding corrections (left and right panels of Fig. 10, respectively, which are occurrence density plots with the colours corresponding to $log_{10} N$, where $N$ is the number of points in the bin). So far, the reflectivity and Doppler velocity measurements have been assumed to be free of noise. Clearly the corrections to NUBF- and wind-shear-induced errors

appear to match very well the actual NUBF- and wind-shear-induced biases. This implies that a very accurate wind field can be constructed from the reflectivity-weighted winds with standard deviations reduced to 0.4 and 0.1 m s$^{-1}$ (see solid black lines in Fig. 9).

However real reflectivity and Doppler velocity data are subject to noise, whose most significant contribution will depend upon the value of the SNR. In the selected configuration a single-pulse SNR of 0 dB corresponds to an echo of -19 dBZ; for a

20 km along track integration (i.e. 200 pulse pairs) the errors due to the pulse pair processing are of the order of 1.5 m s$^{-1}$ at 0 SNR (see Fig. 3 in Illingworth et al. (2018b)).

In order to ascertain whether accurate wind fields can be retrieved from the Doppler radar data, it is necessary to examine how well the vertical reflectivity and Doppler velocity gradients, which enter Eqs. (3-4), can be obtained in the presence of varying levels of noise. The effectiveness of the wind corrections therefore has been studied by injecting noise into the reflectivities and

Doppler velocity fields. This noise has been assumed to take the form of normally distributed fluctuations whose magnitudes are determined by the actual SNR, the spectral width of the Doppler spectrum and the selected $T_{hv}$ (according to formula 6 in Hogan et al. (2004) for the reflectivity noise and formula 15 in Pazmany et al. (1999) for the Doppler velocity noise). Since the addition of noise causes large variations in reflectivity and wind gradients obtained from noisy data, efforts have been made to mitigate any consequent negative effects by applying a running mean to the data along the line of sight (averaging over three

vertical range bins) with the intent of smoothing them before the gradients are computed. The solid red lines in Fig. 9 show the corrected errors after the injection of noise with no averaging. The solid red line for the NUBF-induced error (left panel) shows the correction for all data whereas the solid red line for the wind-shear-induced error (right panel) is for $SNR \geq 16$ dB (see discussion below centred on Fig. 11).





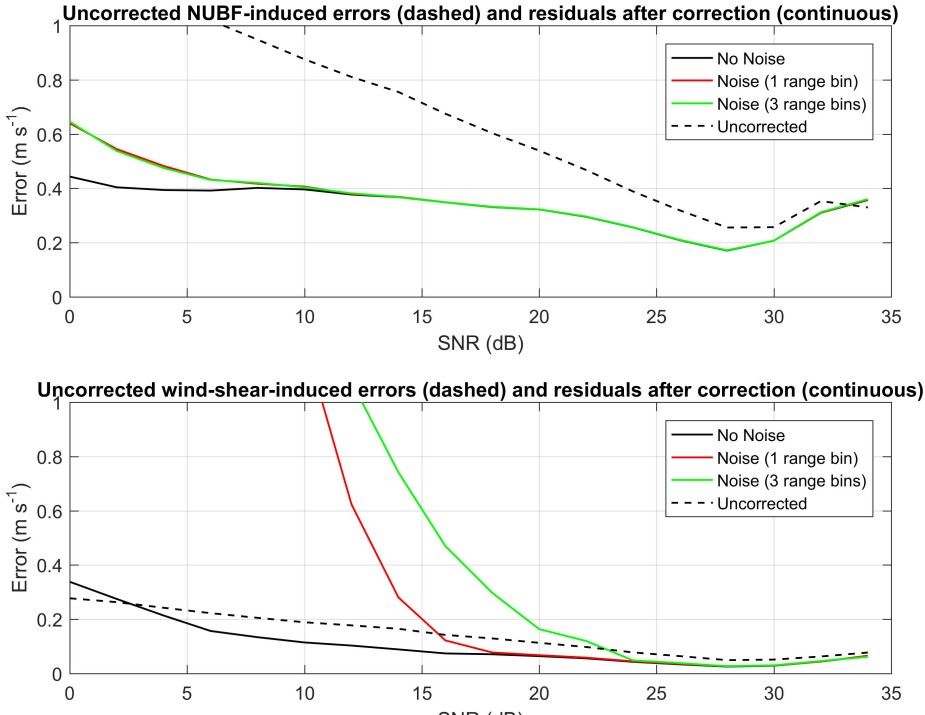

**Figure 11.** The residual NUBF-induced (top) and wind-shear-induced (bottom) errors as a function of the SNR without noise (black solid lines) and with added unaveraged noise (red solid lines) and for a running average over three vertical range bins (green solid lines) for the eight days of CloudSat data. The initial uncorrected error is shown using black dashed lines.

The errors in the wind estimates due to the residual NUBF-induced and wind-shear-induced effects as a function of the SNR are shown in Fig. 11 (top and bottom panels, respectively). The horizontal axis shows the signal-to-noise ratio, with 0 dB corresponding to a reflectivity of -19 dBZ, and the vertical axis shows the standard deviation (error) of the NUBF-induced and wind-shear-induced errors. The corrected (residual) values are shown for the cases of no noise added to the correction (black solid line), and corrections using added noise and a running mean of three vertical range bins of $\tilde{0}.4$ km (red and green solid lines respectively). Also shown are the uncorrected biases where the corrections given by Eq.3 and Eq.4 have not been applied (black dashed lines).

Clearly the corrections for NUBF-induced errors vary little with SNR. Even without averaging, the correction based on Eq. (3) produces results similar to those for the perfect correction without any noise (black solid line), and is only slightly worse for $SNR < 10$ dB. It is worth applying the correction as the error is reduced significantly from its initial uncorrected value (black dashed line). There is an increase in the corrected and uncorrected NUBF-induced error for SNR above about 28





dB, although the error remains below 0.4 m $^{-1}$. This is caused by a tendency for data to deviate away from the one-to-one line (grey dashed line in Fig. 10) at the highest SNR values, towards a line with a lower gradient. This indicates that the correction given by Eq. 3 may be less effective at these highest SNR values. Horanyi et al. (2014) show that the bias must be less than 1 m $^{-1}$ but that higher random errors are acceptable: the errors here documented appear to largely fulfill such conditions.

5      For the case of wind-shear-induced errors, the correction clearly becomes more effective for higher SNR, with the corrected error approaching that for the no-noise case for $SNR \approx 20$ dB (red line). For $SNR < 20$ dB, there is a significant deviation between the noise and no-noise cases. The initial uncorrected wind-shear-induced error is very small for all SNR, and the no-noise correction reduces this already small error quite appreciably. Adding noise clearly affects the corrected value significantly, particularly for $SNR < 20$ dB (red solid line). This increase in the wind-shear-induced residual error is caused by noise-induced variations in the velocity gradient. These noise-induced changes render the correction ineffective at lower SNR. Interestingly, averaging over three vertical range bins makes the correction worse (green solid line) with the corrected value only approaching the no-noise value at about 25 dB SNR. In contrast to the case for NUBF-induced errors, where the correction produces notable improvement for all SNR values, the correction for wind-shear-induced effects will reduce the error significantly only if it is applied for $SNR > 18$ dB.

## 4.2    Cross-talk errors and optimal $T_{hv}$ selection

Simulations of Doppler velocities have been generated using $T_{hv}$ values ranging from 5 to 40 $\mu$s, taking into account the SNR and the strengths of the ghost echoes for 20-km along-track integration. The fraction of profiles for which a "WIVERN forward" configuration is expected to produce winds with accuracy better than 2 m s$^{-1}$ is presented in Fig. 12 for high-latitudes (top), mid-latitudes (middle) and tropical (bottom panel) oceanic conditions. Dashed (continuous) lines correspond to results when the ghosts are (are not) accounted for. Clearly the ghosts only marginally reduce the number of WIVERN measurements with accuracy better than 2 m $^{-1}$. Specifically, the number of measurements with good accuracy is reduced by 1.8%, 0.9% and 0.3% for $T_{hv}$ of 5, 20 and 40 $\mu$s respectively.

The fraction is much lower for the 5 $\mu$s $T_{hv}$ values because at this pair separation a small noise in the phase propagates into a large velocity error. Overall Fig. 12 shows that in the mid-troposphere (3-8 km) "WIVERN forward" would provide a useful measurement 10% of the observation time, but this is expected to be significantly reduced over land at heights near to 2 and 4 km where the bright land surfaces produce ghost echoes for $T_{hv}$ of 20 and 40 $\mu$s, respectively.

The selection of an optimal $T_{hv}$ certainly accounts for the result produced in Fig. 12 but other factors must also be considered:

1. Aliasing must be avoided. A $T_{hv} = 20$ $\mu$s corresponding to a Nyquist velocity $v_N \approx 40$ m s$^{-1}$ is certainly advantageous compared to a $T_{hv} = 40$ $\mu$s corresponding to a $v_N \approx 20$ m s$^{-1}$ when winds in extreme weather events are sought after.

2. Winds in the mid-troposphere are considered to be more important than winds in the lower troposphere. This is because there is a lack of winds in the middle troposphere (see Fig. 9 in Illingworth et al. (2018b)) and because winds in the lower troposphere are more affected by boundary layer dynamics (which are less long-lived and therefore more difficult to assimilate). As a result it is preferable to have the surface ghosts as close to the surface as possible. Here only sea





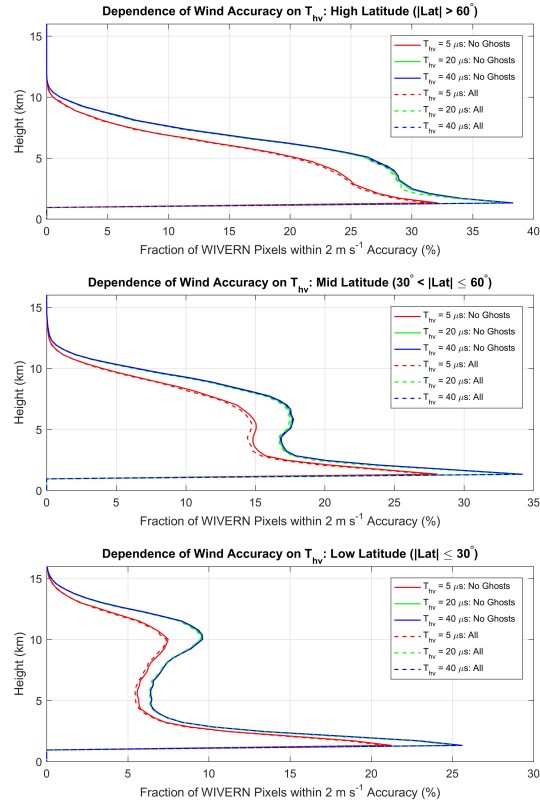

**Figure 12.** The fraction of profiles (including cloudy and non-cloudy situations) where winds at a given height can be derived with an accuracy of 2 m s$^{-1}$ for various $T_{hv}$ values for high-latitudes (top), mid-latitudes (middle) and tropical (bottom panel) oceanic conditions. Dashed (continuous) lines correspond to results when the ghosts are (are not) accounted for. A "WIVERN forward" configuration has been assumed (see Tab. 1) with an along-track integration length of 20 km. The WIVERN profiles have been reconstructed from CloudSat profiles over ocean with $LDR$ and clutter signals reconstructed based on airborne observations of ocean surface returns.

surfaces have been considered; land surfaces are much brighter (Battaglia et al., 2017) and therefore surface ghosts will play a more notable role. Again this favours smaller $T_{hv}$, e.g. the $T_{hv} = 20$ $\mu$s (ghost layer centred at 2.2 km) vs 40 $\mu$s (ghost layer centred at 4.5 km).

3. In the presence of large turbulence the Doppler spectral width can increase and greatly exceed the value predicted by accounting for the satellite motion only (Battaglia and Kollias, 2014a). This decreases the coherency time of the medium and, as a consequence, the optimal $T_{hv}$ value that minimizes the noise error is also reduced (see Fig. 8 in Battaglia and Kollias (2014a)).





## 5 Conclusions and future work

CloudSat observations and Level 2 products have been used in combination with collocated ECMWF wind reanalysis to simulate spaceborne oblique-viewing W-band Doppler observations for radars adopting polarization diversity, which have been recently proposed within the Earth Observation programmes of various space agencies. A specific radar configuration (the "WIVERN", see Tab. 1), recently proposed for the ESA-Earth Explorer 10 call, is analysed in detail in this study. The simulator capitalizes on the fact that the CloudSat W-band Cloud Profiling Radar is in a polar orbit and therefore provides realistic global patterns of the radar reflectivity spatial variability at the W-band frequency range (and, when combined with ECMWF-winds, of how such patterns co-vary with the wind-field), which are key drivers for establishing the performance of a W-band Doppler system on the global scale. The simulator is particularly suited for assessing the relevance of non-uniform beam filling and wind-shear driven errors and of the effectiveness of their corrections based on the estimates of vertical gradients of the reflectivity and wind fields.

For data assimilation properly quantified random wind errors are generally acceptable but biases larger than 1 ms$^{-1}$ cannot be tolerated. Our testing dataset based on roughly 100 CloudSat granules shows that for an instrument looking at $41°$ slant angle and for a 20 km integration both wind shear and NUBF introduce almost unbiased errors (biases of -0.3 and 0 m s$^{-1}$, respectively) with standard deviations of the order of 0.9 and 0.2 m s$^{-1}$. Such errors can be reduced if vertical gradients of reflectivity and of the wind can be estimated via Eqs. (3-4). If vertical gradients are perfectly estimated then the residuals in the Doppler velocities are unbiased and their standard deviations can be reduced to 0.2 and 0.1 m s$^{-1}$. Practically, the corrections for both NUBF-induced and wind-shear-induced errors are effective in producing unbiased velocities (and perform at their best with SNR$> 10$ dB and SNR$> 20$ dB respectively).

The simulator also allows the quantification of the average number of accurate measurements that could be potentially gathered by the Doppler radar for each orbit. This is strongly impacted by the selection of the polarization diversity $H - V$ pulse separation, $T_{hv}$. For the WIVERN slant configuration a selection close to 20 $\mu$s seems to achieve the right balance between maximizing the number of accurate wind measurements (exceeding $10\%$ in the mid-troposphere), and minimizing aliasing effects in the presence of high winds. The presence of cross-talk reduces only marginally the region of measurements with good accuracy.

This study represents the first step towards a more sophisticated development of end-to-end simulators for spaceborne oblique viewing Doppler radars adopting polarization diversity. Further work could improve the simulator along the following guidelines.

1. The CloudSat radar only provides a 2D curtain, and thus does not capture the full 3D structure of clouds. Moreover its vertical (500 m) and horizontal ( $1.4 \times 1.7$ km$^2$) resolutions (Stephens et al., 2008) are not optimal for fully characterizing the spatial variability within the scattering volumes envisaged for future spaceborne W-band Doppler radars. Future scanning radar missions will be able to provide 3D structure of clouds (e.g. Durden et al. (2016)), and are expected to be launched in the next decade (The Decadal Survey, 2017). In the meantime cloud down-scaling algorithms and stochastic





models such as those proposed in e.g. Venema et al. (2010); Barker et al. (2011) could be applied to the CloudSat dataset in order to mimic the natural variability at smaller scales.

2. The effective reflectivity is retrieved for CloudSat profiles only over ocean, where an estimate of the path-integrated attenuation is possible via the surface reference technique (Haynes et al., 2009). The analysis conducted in this paper is therefore limited to ocean cases only. The EarthCARE Cloud Profiling Radar, expected for launch in 2020 (Illingworth et al., 2015), with the inclusion of Doppler capabilities has the potential to enable estimates of rain rate without the need for an estimate of the path-integrated attenuation (Mason et al., 2017). Therefore the reconstruction of effective reflectivity and attenuation profiles should become possible over land as well.

3. The ECMWF wind fields are defined at coarse horizontal scales whereas the effective vertical resolution exceeds 1.5 km so that they generally tend to underestimate the vertical wind shear (Houchi et al., 2010), with mean and median values of wind shears from radiosondes a factor of two larger than ECMWF outputs. Reanalysis at finer resolutions like ERA-5 should mitigate this issue.

*Acknowledgements.* This work was supported in part by the European Space Agency under the activity Doppler Wind Radar Demonstrator (ESA-ESTEC) under Contract 4000114108/15/NL/MP and in part by CEOI-UKSA under Contract RP10G0327E13. The work by Alessandro Battaglia has been supported by the project Radiation and Rainfall funded by the UK National Center for Earth Observation. This research used the ALICE High Performance Computing Facility at the University of Leicester.



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
