# Peer review of "Doppler W-band polarization diversity spaceborne radar simulator for wind studies"

_Atmospheric Measurement Techniques, 2018_

## Referee Comment (RC1) · Anonymous Referee #1 · 18 Jul 2018

This article provides a nice and thorough description of the expected performance and added-value of a scanning spaceborne Doppler radar with polarization diversity. The article describes the ability of such an instrument to retrieve atmospheric winds from Doppler measurements at W band. The originality of this study is that it considers the scanning operation of a Doppler radar from space, whereas many such studies have focused on the nadir-looking case (with EarthCARE in mind). For scanning radars, error sources such as NUBF become particularly important. The article is well written and the Authors do a good job of reviewing the existing literature and establishing a niche to be addressed. Another strength of this article lies in the use of spaceborne (Cloud-Sat granules), reanalysis (wind fields) and field-campaign (LDR climatology) data for the simulations. The sensitivity analysis places the results of this study in a wider and

more realistic context. Three error sources are investigated in detail, viz. NUBF, wind shear and cross-talk between orthogonal polarization channels. Analytical formulae (Eqs 3,4)are provided to correct for the first two types of Doppler errors. Results of the simulations indicate a high skill of the NUBF correction and of the shear correction (albeit only for larger SNR).

For all the aforementioned reasons, this article is particularly relevant as the scientific community designs future spaceborne clouds and precipitation radars. I therefore recommend it for publication once the following points will have been addressed.

see details in attached document.

- A point made by the Authors is the limited amount of spatial variability of the input data (due to the smearing and coarse horizontal sampling of satellite and reanalysis data), and thereby a possible under-estimation of NUBF and wind shear errors. Since, NUBF and wind shear biases are mostly sub-footprint effects, I was wondering if there was any merit in spatially interpolating (if possible with an order higher than linear) the input data (satellite, ECMWF) to a finer resolution, before computing the Doppler velocity errors?

- P9L10-13: For attenuation, do the Authors 1) correct for attenuation in the CloudSat viewing configuration (using the 2C-RAIN products?), 2) generate unattenuated Z in the WIVERN look direction (and resolution), and 3) add attenuation in the WIVERN viewing direction?  o If not, how much of a limiting factor is it for the realism of your simulations, especially for large off-nadir look angles? o If so, then regions with invalid radar data aloft (due to attenuation or multiple scattering) would invalidate radar data in their "shadow" to the surface (along the viewing direction). This limits the amount of radar data available for the statistical analysis. Could you assess/comment on the penalty incurred by this effect?

- P12L7-9: Wouldn't the dominant contribution come from the vertical only if the look angle exceed 45 degrees? It seems to me that more than a "dominant factor", the key

here is that, for non-nadir look angles, the gradient in the direction orthogonal to the Boresight ("eta") becomes correlated to the vertical gradient.

- P15L22-33: For a pulse-pair radar, the noisiness would be injected when building the (I,Q) voltage samples, and this noisiness would affect both reflectivity and velocity (Approach described in Zrnic 1975 or Sirmans and Baumgartner 1975). Is your addition of noise to the WIVERN Doppler data consistent with the (I,Q)-based approach?

- Do the Authors have recommendations for a better correction of wind-shear-induced errors over a wider range of SNR values?

Please also note the supplement to this comment:
https://www.atmos-meas-tech-discuss.net/amt-2018-184/amt-2018-184-RC1-supplement.pdf

[Figure]

**Supplement:**

This article provides a nice and thorough description of the expected performance and added-value of a scanning spaceborne Doppler radar with polarization diversity. The article describes the ability of such an instrument to retrieve atmospheric winds from Doppler measurements at W band. The originality of this study is that it considers the scanning operation of a Doppler radar from space, whereas many such studies have focused on the nadir-looking case (with EarthCARE in mind). For scanning radars, error sources such as NUBF become particularly important.

The article is well written and the Authors do a good job of reviewing the existing literature and establishing a niche to be addressed. Another strength of this article lies in the use of spaceborne (CloudSat granules), reanalysis (wind fields) and field-campaign (LDR climatology) data for the simulations. The sensitivity analysis places the results of this study in a wider and more realistic context.

Three error sources are investigated in detail, viz. NUBF, wind shear and cross-talk between orthogonal polarization channels. Analytical formulae (Eqs 3,4)are provided to correct for the first two types of Doppler errors. Results of the simulations indicate a high skill of the NUBF correction and of the shear correction (albeit only for larger SNR).

For all the aforementioned reasons, this article is particularly relevant as the scientific community designs future spaceborne clouds and precipitation radars. I therefore recommend it for publication once the following points will have been addressed.

Detailed comments and suggestions (*technical questions in italic*)

- My biggest editorial comment is that the readability of the article could be significantly improved by revising its punctuation (*please, include commas where necessary*). This would really help in conveying the message across without the reader having the read sentences multiple times.

- *A point made by the Authors is the limited amount of spatial variability of the input data (due to the smearing and coarse horizontal sampling of satellite and reanalysis data), and thereby a possible under-estimation of NUBF and wind shear errors. Since, NUBF and wind shear biases are mostly sub-footprint effects, I was wondering if there was any merit in spatially interpolating (if possible with an order higher than linear) the input data (satellite, ECMWF) to a finer resolution, before computing the Doppler velocity errors?*

- **Abstract**
  o Page 1, Line 24 (P1L24): "enable a full capture…"?

- **Introduction**
  o P2L5: "water vapour" …
  o P2L12: "In this approach, active…"
  o P4L26: "…100 orbits…" please remove the tilde superscript.

- **Section 2**
  o P4L30: "… Doppler radars, two …"

- **Section 3**
  - P5L22: "Therefore, we…"
  - Titles of Figs 4,5,6: Please use a single notation for WIVERN (or Wivern) throughout the article;
  - P6L3: "…Cape Verde Islands…"
  - P9L1: "In order To simulate…"
  - P9L9: "In order To produce…radars, two aspects…"
  - *P9L10-13: For attenuation, do the Authors 1) correct for attenuation in the CloudSat viewing configuration (using the 2C-RAIN products?), 2) generate unattenuated Z in the WIVERN look direction (and resolution), and 3) add attenuation in the WIVERN viewing direction?*
    - *If not, how much of a limiting factor is it for the realism of your simulations, especially for large off-nadir look angles?*
    - *If so, then regions with invalid radar data aloft (due to attenuation or multiple scattering) would invalidate radar data in their "shadow" to the surface (along the viewing direction). This limits the amount of radar data available for the statistical analysis. Could you assess/comment on the penalty incurred by this effect?*
  - Fig6 legend: "… 40m/s)."
  - P10L9: "Here, we… sources of errors, which …"
  - P10L14: "In fact, the…"
  - P10L14-15: Please clarify where the shear comes from: To the best of my understanding, it is the component of the spacecraft velocity along the line-of-sight that causes the shear in all none-nadir look directions.
  - P11L1: Which "black arrow" are you referring to? There are quite a few in Fig.3…
  - P11L4: "…size, this … gradient, this…"
  - P11L5: Please specify in which figure, we can find the volumes 11 and 6.
  - P12L3: "…radars, notional…"
  - P12L5: "…Similarly, …"
  - *P12L7-9: Wouldn't the dominant contribution come from the vertical only if the look angle exceed 45 degrees? It seems to me that more than a "dominant factor", the key here is that, for non-nadir look angles, the gradient in the direction orthogonal to the Boresight ("eta") becomes correlated to the vertical gradient. Please clarify.*
  - P13L1: shouldn't it be "antenna reflectivity-weighted"?
  - P13L19: "…dB, the.."

- **Section 4**
  - *P15L22-33: For a pulse-pair radar, the noisiness would be injected when building the (I,Q) voltage samples, and this noisiness would affect both reflectivity and velocity (Approach described in Zrnic 1975 or Sirmans and Baumgartner 1975). Is your addition of noise to the WIVERN Doppler data consistent with the (I,Q)-based approach?*

- o *P16, Fig 11: Errors reported in terms of standard deviation, wouldn't it be better to express them in terms of RMSE, which would account for effects of biases; That would maybe also help the reader understand Fig.11 …*
- o *Do the Authors have recommendations for a better correction of wind-shear-induced errors over a wider range of SNR values?*
- o P16L5: "…0.4 km…" please remove the tilde superscript.
- o P17L4: "1m/s… errors documented here"
- o P17L20: "Clearly,…"
- o P17L24: "Overall,…"
- o P17L25: "near to .. 4 km": Do you mean "around 2 and around 4 km", or, "between 2 and 4 km of altitude"? Please clarify.
- o Fig12: Please use the same horizontal-axis limits to ease comparisons between the subfigures.
- o P18L2: "Again, this… e.g.  $T_{hv}$…"

- **Section 5**
  - o P19L12: "… assimilation, properly…"
  - o P19L14: "… integration, both…"
  - o P19L24: "… meantime, cloud…"

- **References**
  - o P22L1: "… WIVERN: A new…"
  - o P23L7: "… Pawswson, C. Reynoldsldslds": Please revise

---

## Referee Comment (RC2) · Anonymous Referee #2 · 31 Jul 2018

[referee-annotated manuscript omitted]

---

## Author Comment (AC1) · 2 Aug 2018

Reply to Reviewer 1

Thanks for the thorough review! We have corrected all the minor issues as suggested by the reviewer (they appear in the revised version attached at the end).

Here we provide a reply to the major points raised by the reviewer (highlighted in red).

A point made by the Authors is the limited amount of spatial variability of the input data (due to the smearing and coarse horizontal sampling of satellite and reanalysis data), and thereby a possible under-estimation of NUBF and wind shear errors. Since, NUBF and wind shear biases are mostly sub-footprint effects, I was wondering if there was any merit in spatially interpolating (if possible with an order higher than linear) the input data (satellite, ECMWF) to a finer resolution, before computing the Doppler velocity errors?

The reviewer is right in pinpointing at the sub-footprint variability as the key source of NUBD and wind shear errors. Part of this is certainly captured by the variability in our datasets but as mentioned in our conclusions (point 1 page 19) it is not trivial to downscale reflectivity and wind fields to a finer resolution.   This is left for future studies.

P9L10-13: For attenuation, do the Authors 1) correct for attenuation in the CloudSat viewing configuration (using the 2C-RAIN products?), 2) generate unattenuated Z in the WIVERN look direction (and resolution), and 3) add attenuation in the WIVERN viewing direction?

- If not, how much of a limiting factor is it for the realism of your simulations, especially for large off-nadir look angles?
- If so, then regions with invalid radar data aloft (due to attenuation or multiple scattering) would invalidate radar data in their "shadow" to the surface (along the viewing direction). This limits the amount of radar data available for the statistical analysis. Could you assess/comment on the penalty incurred by this effect?

We do correct for attenuation as the reviewer suggests. Indeed the reason for using the 2C-RAIN product is to get reconstructions of profiles of unattenuated reflectivities and of attenuation that can be used for reconstructing the view at 41 degree incidence angle. Text at page 9 has been modified. The 2C-RAIN product does not produce a result in presence of strong attenuation and/or multiple scattering. For these areas we assume that a shadow region will propagate from an altitude equal to the freezing level height +2km. These shadowed regions are those grey shaded in Fig6. This represents less than 5% of the CloudSat data.

P12L7-9: Wouldn't the dominant contribution come from the vertical only if the look angle exceed 45 degrees? It seems to me that more than a "dominant factor", the key here is that, for non-nadir look angles, the gradient in the direction orthogonal to the Boresight ("eta") becomes correlated to the vertical gradient. Please clarify.

Well, generally speaking, vertical gradients of reflectivities are much larger than horizontal ones. As a result, even if the incidence angle is not exceeding 45degrees, the vertical reflectivity gradients have a larger impact onto NUBF effects.  We have rephrased the statement to make this clear.

P15L22-33: For a pulse-pair radar, the noisiness would be injected when building the (I,Q) voltage samples, and this noisiness would affect both reflectivity and velocity (Approach described in Zrnic 1975 or Sirmans and Baumgartner 1975). Is your addition of noise to the WIVERN Doppler data consistent with the (I,Q)-based approach?

Yes the procedure is similar to the one for a pulse-pair radar, so indeed the noisiness affects both reflectivities and velocities. We realised that in the current version the left panel of Fig.6 represents indeed the 5-km ideal reflectivity field (i.e. the one computed from the CloudSat reflectivities before injecting the noise appropriate for a system with single pulse sensitivity of -19 dBZ). In the revised version we will include the 20-km reflectivity field after noise subtraction (see attached). Note that in this situation we can get reflectivities down to -32.5 dBZ because 20 km integration corresponds to 520 pulses.

[Figure]

We will put this figure in the updated version.

*P16, Fig 11: Errors reported in terms of standard deviation, wouldn't it be better to express them in terms of RMSE, which would account for effects of biases; That would maybe also help the reader understand Fig.11 …*

*The figure has been changed and everything has been plotted in terms of RMSE. The text commenting the figure has been changed. .*

○ *Do the Authors have recommendations for a better correction of wind-shear-induced errors over a wider range of SNR values?*

*No we do not see any method to correct for it. But such error remains small compared to the other errors involved in such measurement.*

*A revised version with all correction in place is attached.*

[revised manuscript text omitted]

---

## Author Comment (AC2) · 2 Aug 2018

See file attached, hopefully not damaged this time!

Please also note the supplement to this comment:
https://www.atmos-meas-tech-discuss.net/amt-2018-184/amt-2018-184-AC2-
supplement.pdf

---

## Author Comment (AC3) · 2 Aug 2018

We thanks the reviewer for the corrections. We have implemented them in the new version of the paper that can be found at the end of the reply to reviewer 1.

---

## Author Response (AR2)

**Reply to Reviewer 1**

Thanks for the thorough review. We have corrected all the minor issues as suggested by both reviewers. The corrections (and the previous version) appear in the revised version attached at the end of this file.

Here we provide a reply to the major points raised by the reviewer 1 (highlighted in red) with our response (blue) and our changes in the manuscript (green, with page and lines referring to the version attached at the end of this file).

A point made by the Authors is the limited amount of spatial variability of the input data (due to the smearing and coarse horizontal sampling of satellite and reanalysis data), and thereby a possible under-estimation of NUBF and wind shear errors. Since, NUBF and wind shear biases are mostly sub-footprint effects, I was wondering if there was any merit in spatially interpolating (if possible with an order higher than linear) the input data (satellite, ECMWF) to a finer resolution, before computing the Doppler velocity errors?

The reviewer is right in pinpointing at the sub-footprint variability as the key source of NUBD and wind shear errors. Part of this is certainly captured by the variability in our datasets but as mentioned in our conclusions (point 1 page 20) it is not trivial to downscale reflectivity and wind fields to a finer resolution (which spatial interpolation should be selected?). This is left for future studies.

No change done to the manuscript.

P9L10-13: For attenuation, do the Authors 1) correct for attenuation in the CloudSat viewing configuration (using the 2C-RAIN products?), 2) generate unattenuated Z in the WIVERN look direction (and resolution), and 3) add attenuation in the WIVERN viewing direction?

- If not, how much of a limiting factor is it for the realism of your simulations, especially for large off-nadir look angles?
- If so, then regions with invalid radar data aloft (due to attenuation or multiple scattering) would invalidate radar data in their "shadow" to the surface (along the viewing direction). This limits the amount of radar data available for the statistical analysis. Could you assess/comment on the penalty incurred by this effect?

We do correct for attenuation as the reviewer suggests. Indeed the reason for using the 2C-RAIN product is to get reconstructions of profiles of unattenuated reflectivities and of attenuation that can be used for reconstructing the view at 41 degree incidence angle. The 2C-RAIN product does not produce a result in presence of strong attenuation and/or multiple scattering. For these areas we assume that a shadow region will propagate from an altitude equal to the freezing level height +2km. These shadowed regions are those grey shaded in Fig6. This represents less than 5% of the CloudSat data.

Text at the bottom of page 8 and top of page 9 and line 9-10 at page 11 has been modified to clarify this aspect. Figure 4 and Figure 6 have been replotted to show the regions where radar data are invalid due to the presence of attenuation/multiple scattering, showing also the impact of integration.

P12L7-9: Wouldn't the dominant contribution come from the vertical only if the look angle exceed 45 degrees? It seems to me that more than a "dominant factor", the key here is that, for non-nadir look angles, the gradient in the direction orthogonal to the Boresight ("eta") becomes correlated to the vertical gradient. Please clarify.

Well, generally speaking, vertical gradients of reflectivities are much larger than horizontal ones. As a result, even if the incidence angle is not exceeding 45degrees, the vertical reflectivity gradients have a larger impact onto NUBF effects.

We have rephrased the statement to make this clear (page 13 line 8-10).

P15L22-33: For a pulse-pair radar, the noisiness would be injected when building the (I,Q) voltage samples, and this noisiness would affect both reflectivity and velocity (Approach described in Zrnic 1975 or Sirmans and Baumgartner 1975). Is your addition of noise to the WIVERN Doppler data consistent with the (I,Q)-based approach?

Yes the procedure is similar to the one for a pulse-pair radar, so indeed the noisiness affects both reflectivities and velocities. We realised that in the current version the left panel of Fig.6 represents indeed the 5-km ideal reflectivity field (i.e. the one computed from the CloudSat reflectivities before injecting the noise appropriate for a system with single pulse sensitivity of -19 dBZ).

In the revised version we have included the 20-km reflectivity field after noise subtraction (see attached). Note that in this situation we can get reflectivities down to -32.5 dBZ because 20 km integration corresponds to 520 pulses.

[Figure]

This figure has been included in the updated version.

P16, Fig 11: Errors reported in terms of standard deviation, wouldn't it be better to express them in terms of RMSE, which would account for effects of biases; That would maybe also help the reader understand Fig.1.

Agreed

The figure has been changed and everything has been plotted in terms of RMSE (the figure is not much different). The text commenting the figure has been changed (see new text at page 16).

*Do the Authors have recommendations for a better correction of wind-shear-induced errors over a wider range of SNR values?*
*No we do not see any method to correct for it. But such error remains small compared to the other errors involved in such measurement.*
*No change done.*

**Reply to Reviewer 2**

All minor revisions recommended by reviewer 2 have been implemented in the new document.

[revised manuscript text omitted]